# New and Developing Therapies in Spinal Muscular Atrophy: From Genotype to Phenotype to Treatment and Where Do We Stand?

**DOI:** 10.3390/ijms21093297

**Published:** 2020-05-07

**Authors:** Tai-Heng Chen

**Affiliations:** 1Section of Neurobiology, Department of Biological Sciences, University of Southern California, Los Angeles, CA 90089, USA; taihen@kmu.edu.tw; 2Division of Pediatric Emergency, Department of Pediatrics, Kaohsiung Medical University Hospital, Kaohsiung Medical University, Kaohsiung 80708, Taiwan; 3Ph.D. Program in Translational Medicine, Graduate Institute of Clinical Medicine, Kaohsiung Medical University and Academia Sinica, Taipei 11529, Taiwan

**Keywords:** spinal muscular atrophy, survival motor neuron protein, novel therapy, clinical care

## Abstract

Spinal muscular atrophy (SMA) is a congenital neuromuscular disorder characterized by motor neuron loss, resulting in progressive weakness. SMA is notable in the health care community because it accounts for the most common cause of infant death resulting from a genetic defect. SMA is caused by low levels of the survival motor neuron protein (SMN) resulting from *SMN1* gene mutations or deletions. However, patients always harbor various copies of *SMN2*, an almost identical but functionally deficient copy of the gene. A genotype–phenotype correlation suggests that *SMN2* is a potent disease modifier for SMA, which also represents the primary target for potential therapies. Increasing comprehension of SMA pathophysiology, including the characterization of *SMN1* and *SMN2* genes and SMN protein functions, has led to the development of multiple therapeutic approaches. Until the end of 2016, no cure was available for SMA, and management consisted of supportive measures. Two breakthrough SMN-targeted treatments, either using antisense oligonucleotides (ASOs) or virus-mediated gene therapy, have recently been approved. These two novel therapeutics have a common objective: to increase the production of SMN protein in MNs and thereby improve motor function and survival. However, neither therapy currently provides a complete cure. Treating patients with SMA brings new responsibilities and unique dilemmas. As SMA is such a devastating disease, it is reasonable to assume that a unique therapeutic solution may not be sufficient. Current approaches under clinical investigation differ in administration routes, frequency of dosing, intrathecal versus systemic delivery, and mechanisms of action. Besides, emerging clinical trials evaluating the efficacy of either SMN-dependent or SMN-independent approaches are ongoing. This review aims to address the different knowledge gaps between genotype, phenotypes, and potential therapeutics.

## 1. Introduction

Spinal muscular atrophy (SMA) is an autosomal recessive neurodegenerative disease characterized by devastating muscular wasting caused by the progressive degeneration of spinal motor neurons (MNs). Although recognized as a rare disease with an estimated global incidence of ~1/10,000 live births, SMA has been regarded as a noteworthy health issue because it is not only the second most common autosomal recessive inherited disorder, but is also the most common monogenic disease leading to infant mortality [1,2]. The carrier frequency of SMA varies from 1 in 38 to 1 in 72, depending on the racial group, with the pan-ethnic average being 1 in 54 [3].

Figure 1 illustrates the genetic pathomechanism of SMA. All patients with SMA have an insufficient amount of a survival motor neuron (SMN) protein, which is encoded by two highly homogenous genes, survival of motor neuron 1 *(SMN1)* and its copy gene, *SMN2*. *SMN2* is differentiated from *SMN1* by one single nucleotide variant (C→T) in exon 7. This critical difference results in the preferential exclusion of exon 7 from most (~90%) *SMN2* transcripts, termed SMN△7, which translates into truncated and unstable SMN protein. As a consequence, SMN2 can only generate ~10% of full-length (FL) SMN mRNAs and their product-functional SMN proteins (Figure 1A). While these FL-SMN2 transcripts can partially compensate for the loss of SMN1, it is reasoned that retained SMN2 copy numbers of patients determine the phenotypic severity (Figure 1B). However, such a phenotype–genotype correlation is not absolute, as recent studies have indicated that additional cellular mechanisms (e.g., positive or negative disease modifiers) might also involve the modulation of SMA clinical severity. For example, rare SMN2 variants (c.859G > C), as well as independent modifiers such as plastin 3 or neurocalcin delta, can further influence the disease severity [4,5,6]. In brief, the loss of the SMN1 gene leads to SMA, whose severity is partially modified by various copies of SMN2.

Understanding SMN protein functions and mechanisms of action in subcellular contexts may elucidate potential pathways for therapeutic intervention. SMN is a multifunctional protein that is ubiquitously expressed in most somatic cells [7]. The most appreciated canonical role of SMN is to serve as an essential component of small nuclear ribonucleoproteins (snRNPs) that form spliceosomes to process the pre-mRNA splicing [8,9]. Studies on SMA animal models have revealed a direct correlation between the ability to assemble snRNPs and SMA phenotypes [10]. SMN is also involved in DNA repair and mRNA transportation along MN axons [11]. However, the multifaceted roles of SMN protein are still under investigation, and it is unclear how a deficiency in ubiquitously expressed SMN can selectively cause the dramatic MN degeneration. The cell autonomous effects related to deficient SMN are responsible for the MNs’ degeneration. However, this does not account for the full SMA phenotype, implicating not only the dysfunction of neural networks but other non-neuronal cell types involved in the disease process [12,13]. For example, recent studies indicate that the MN survival and functionality of SMA animal and cellular models is highly dependent on glial cells, which play an important role in neuronal communication and neuroinflammation [14,15]. These findings imply that SMA could also be a neuroinflammatory disease.

## 2. Clinical Characteristics of SMA

### 2.1. SMA Phenotypes and Classifications

SMA is classified into three main phenotypes based on age at the onset of symptoms/signs, and the highest motor function achieved [3,6,16]. However, some patients with SMA are outliers on either end of the phenotypic spectrum. Subclassification has also been proposed in SMA types 1 and 3, and sometimes in the type 2 phenotype (Table 1 and Figure 1B). It should be kept in mind that regardless of the type of SMA, severity may vary and change over time, because the disease is a continuum.

At one end of the spectrum, patients with type 0 SMA (categorized as type 1A by some authors) are usually associated with prenatal onset of signs such as a history of decreased fetal movements [17]. These rare cases usually present with arthrogryposis multiplex congenita and have profound hypotonia and respiratory distress soon after birth. Life expectancy is hugely reduced, and if untreated, most cases are unable to survive beyond one month of age [1,18]. 

Type 1 SMA patients account for more than 50% of the incidence of SMA. These patients are called non-sitters, who never achieve major motor developmental milestones such as sitting, standing, and walking in their lifetime. Notably, congenital heart defect is also a feature of severe SMA phenotype, especially in SMA types 0 and 1 [19]. If left untreated, most cases of mortality within the first two years of life are attributable to respiratory muscle dysfunction, with a median survival of 13.5 months [20].

Patients with type 2 SMA (Dubowitz’s disease) are sitters who initially present with a delay in reaching developmental milestones for gross motor skills. Although these patients can maintain a sitting position unaided (“sitters”) and a few can even stand with leg braces, none can walk independently. Kyphoscoliosis usually develops, complicated with restrictive lung disease if there are no orthopedic interventions. Cough and ability to clear airway secretions are usually progressively compromised. A majority of patients with type 2 SMA can survive into adulthood. However, after entering their adolescence, these patients usually require aggressive supportive management, especially regarding gastrointestinal and respiratory complications [21,22].

Patients with type 3 SMA (Kugelberg–Welander disease) are walkers who are able to stand unsupported and walk independently during their early life. These patients usually show profound symptom heterogeneity, so they are often misdiagnosed with myopathy or muscular dystrophy. The distribution of weakness is similar to that seen in patients with types 1 and 2 SMA, but the progression of weakness is much slower; some patients may eventually become wheelchair dependent at middle age [23]. At the other end of the spectrum is the mildest adult-onset form, known as type 4 SMA, where individuals present onset symptoms—usually a weakness of lower extremities—after the second decade. Type 4 patients have a good prognosis with ambulation into adulthood and a mostly average life span [24]. 

### 2.2. The Implication of Phenotypic Classification in SMA Clinical Trials

Efforts to better understand the natural history and define outcome measures pave the way for the readiness of SMA trials [25,26]. Initially, the phenotype–genotype correlation encouraged the application of *SMN2* copies as a criterion for patient enrollment in clinical trials [27]. However, discordant cases certainly exist, and the prediction of phenotype solely through *SMN2* copies is not always accurate in individual cases [6]. Except for some disease modifiers, several prognostic factors have been identified [18]. Although the severity-based classification has clinical advantages, it is not always adequate to provide prognostic information or to facilitate the stratification of SMA patients. Rather than the SMA phenotype, the ambulant status may be more relevant to the trajectory of disease progression [28]. This approach acknowledges the SMA phenotypes as a continuum and focuses on the current functional status and the response to therapy. Furthermore, pulmonary function assessment may better reflect disease state than muscle strength [29]. Nevertheless, repeat evaluations are imperative before assigning a patient to a specific SMA type. Particularly in patients with SMA types 2 and 3, the onset, time course, and extent of MN loss has not been well established, yet are vital in determining whether there is a specific therapeutic window for these patients with milder phenotypes. 

## 3. Impacts of Evolving Supportive Care in SMA Therapeutic Era

Advances in drug development are likely to impact the natural history of and care methods for patients with SMA [28,30]. An immediate benefit of multidisciplinary care and care coordination for SMA patients has been the development and distribution of standard-of-care (SOC) recommendations. In 2007, a consensus statement for the standard of care in SMA was released by a multidisciplinary team regarding the current best advice for the management of patients with SMA [31]. With the advent of disease-modifying treatment that became available in early 2017, a new two-part SOC consensus paper was published in 2018 [21,22]. The multidisciplinary team should and may include a variety of medical specialties that ideally follow up both the as-yet untreated patients as part of providing a SOC and patients that undergo specific therapies (Figure 2). As patients following SOC guidelines can receive same-level care to reduce the variability of disease progression, it is particularly crucial for those currently participating in clinical trials. Experts recommend that SMA patients on trial should be paired with adherence to SOC provided a core facility of a multidisciplinary care team [16,28]. As such, revolving SOC guidelines for SMA have assumed a more substantial role of defining what is meant by optimal or even required care.

In the therapeutic era, we reasonably expect that type 1 SMA patients will likely transition into less severe types 3 and 4 once treated, giving them a more extended or average lifespan. It remains unclear whether persistent interventions will be required, and a complete long-term reversal of symptoms will be attained. Unfortunately, because there is a paucity of studies investigating the support and medical needs of type 4 SMA patients (and soon the treated patients), it is unknown whether such lifespan extension will reveal new, previously unknown, comorbidities that could arise with age in this new, modified SMA affected population. In parallel with pre-clinical advances, continued evolution in multidisciplinary care with technological advances should be pursued, particularly for those with milder phenotypes after disease-modifying therapy. 

## 4. Recent Advances in Innovative Therapeutic Approaches for SMA: Focusing SMN and Beyond

In general, the therapeutic strategies in SMA can be categorized either as SMN-dependent therapies or as SMN-independent therapies, which can be subsequently divided into eight different therapeutic approaches (Figure 3). Deletion or mutation of *SMN1* is partially compensated by limited expression of SMN protein produced by variable *SMN2* copies, which provide a therapeutic target [32]. With the proof-of-concept, the initial approaches mostly aim to target *SMN2* in the treatment of SMA [33,34]. However, increasing evidence extend the pathogenic effect of SMN deficiency beyond MNs to include additional cells both within and outside the CNS, whereby numerous peripheral organs and non-neuronal tissues (e.g., cardiovascular system, immune system, gastrointestinal tract, and kidneys) have demonstrated pathological changes in pre-clinical models and patients [12,30,35,36,37,38]. 

We summarize the updated information of pre-clinical and clinical trials for potential therapeutic agents in Table 2. The precise characterization of SMN-dependent and SMN-independent pathways that are both affected and underlying the disease remains a critical aspect of developing therapeutic approaches for SMA. Among different approaches, strategies with the most promising clinical data for SMA have been achieved through upregulating FL *SMN2* production by modulating splicing or replacing functional exogenous *SMN1* gene via a viral vector [39,40]. These two therapies have been officially approved within the past two years. In parallel with the treatment pipeline of SMN-dependent approaches, neuroprotective agents, myostatin inhibitors, skeletal muscle troponin activators, and stem cell therapy are examples of adjunctive SMN-independent therapies [41]. Importantly, recent breakthroughs in novel therapies for SMA may also inspire similar approaches for other genetic motor neuron diseases. For example, similar therapeutic strategies are proposed to applied to spinal muscular atrophy with respiratory distress type 1 (SMARD1) caused by *IGHMBP2* gene mutation is a non-5q SMA, which is the second most common motor neuron disease of infancy following SMA [42].

In the following, we discuss how the current SMN-targeting compounds that are presently in clinical trials inform the potential development of treatments aimed at non-SMN targets in non-CNS tissues. We also discuss the critical role of supportive care methods in the era of SMA therapeutics.

The therapeutic approaches for SMA are generally categorized into SMN-dependent and SMN-independent therapies, which can be further divided into four branches of development, respectively. The yellow circle in Figure 3 indicating *SMN1* gene replacement therapy of the SMN-dependent pathway highlights its difference from the other three therapies in the SMN-dependent category, which mainly target *SMN2*. The dashed lines of the outer rims connecting the SMN-dependent and SMN-independent approaches imply the potential for combinatory effect as a “cocktail therapy” for SMA.

## 5. SMN-Dependent Therapies for SMA

Because of the presence of *SMN2*, a well-validated target for therapeutic interventions, researchers have regarded SMA as a paradigm of “translational” disease. As a proof-of-concept, the most tempting approach in treating SMA is to upregulate *SMN2,* retaining in all patients, to function as the missing *SMN1,* either by activating the *SMN2* gene or by modulation of *SMN2* splicing [1,3,6,30]. As shown in Table 2, this idea prompted investigations into the upregulation of *SMN2* transcription by activating promoter, enhancing exon 7 inclusion, introducing *SMN1* gene via a viral vector, modulating SMN protein translation, and preventing SMN protein degradation.

### 5.1. Previous SMN-Dependent Trials with Indefinable Outcomes

Histones are core proteins of chromatin that play a role in the epigenetic regulation of gene expression via their acetylation status. Early studies investigated the therapeutic potential of histone deacetylase inhibitors (HDACIs) and demonstrated their ability to increase *SMN2* transcription through the modification of chromatin structure in vitro and in vivo SMA models [34,43]. Several potential HDACIs have been proposed to benefit SMA, including valproic acid, phenylbutyrate, and trichostatin A, which have been demonstrated to activate the *SMN2* promotor, driving increased FL SMN [41,44]. However, regardless of any putative effect observed in vitro, no beneficial effect of HDACIs has carried over to clinical trials [45]. Otherwise, it seems that HDACIs are not exclusively specific to SMN, thus leading to a potential for side effects and potential dose limitations [43,46].

Besides histone acetylation, several molecular mechanisms (e.g., histone phosphorylation, ubiquitination, and DNA methylation) are known to affect *SMN2* expression [47]. Hydroxyurea, an FDA-approved non-HDACI agent, was found to increase the amount of FL SMN transcript and protein in vitro [48]. However, a small pilot trial on types 2 and 3 SMA showed no statistically significant benefit [49], followed by negative results of a further placebo-controlled trial [50].

Albuterol is a β-adrenergic agonist which has been shown to increase FL SMN transcript levels in vivo [51]. Two open-label trials of albuterol showed increased FL-SMN transcripts and improvements of motor function in types 2 and 3 SMA patients [52,53]. However, no data of further placebo-controlled trials are available in order to validate the benefit of albuterol in clinical practice for SMA [41].

Despite promising pre-clinical data, there are negative results following clinical trials of valproic acid combined with acetyl-L-carnitine, phenylbutyrate, hydroxyurea, and somatotropin. Their further development was discontinued, yet albuterol is still broadly prescribed off-label. However, these negative studies have informed clinical trial design, validated the reliability and feasibility of specific outcome measures, and highlighted the importance of patient stratification [30,54].

### 5.2. Nusinersen: The First Approved Splicing-Modify Therapy for SMA

Researchers first discovered an intronic splicing silencer N1 (ISS-N1) sequence in intron 7 of *SMN2* involved in mRNA exon 7 skipping [55]. Inhibiting ISS-N1 motifs by antisense oligonucleotides (ASOs) was shown to enhance *SMN2*-mRNA exon 7 inclusion, and improved SMA phenotypes [56,57]. In 2011, a phase 1 trial of nusinersen, one of the ASOs with the greatest potential, demonstrated safety and effectiveness in SMA patients through delivery into the cerebrospinal fluid (CSF) space [58]. The subsequent phase 3 placebo-controlled trial (ENDEAR) showed a significant improvement in motor function and survival in treated infants with type 1 SMA [40]. Nusinersen was approved by the Food and Drug Administration (FDA) in late December 2016, and by the European Medicines Agency in June 2017. 

To date, more than 8000 SMA patients have undergone nusinersen therapy worldwide [6]. However, besides a high price tag of $750,000 for the first year of treatment, questions about its long-term efficacy abound, and there are some restrictions to the use of nusinersen. First, preclinical studies suggest a discrete time-window in neuromuscular development when increasing SMN levels are most effective [59]. The data from human trials also support the importance of a therapeutic window for a SMN-augmented treatment [60]. Unfortunately, SMA newborn screening programs have not yet been extensively performed worldwide or even nationwide [61,62]. On the other hand, patients with later-onset type 2 SMA showed significant motor improvement after treatment [63]; however, whether the long-term effect will be seen when treatment is initiated in the later SMA phase with slow decline is still unclear [64].

Second, because nusinersen cannot penetrate the blood–brain barrier (BBB), beyond which the targeted rescuing MNs lie, there is unfortunately no practical alternative to periodic intrathecal administration. The risks of performing lumbar puncture in SMA patients include exacerbating respiratory compromise related to knee-to-chest flexion posture during the procedure, headache, and CSF leakage. Without modern imaging assistance, repeated intrathecal injections can present challenges in some chronic SMA patients with significant scoliosis [65].

Third, the direct delivery of nusinersen into the spine restricts SMN upregulation only at the CNS; however, there is emerging evidence that SMN also plays a vital role in peripheral tissues [66,67]. Previous studies also demonstrated that peripheral SMN restoration compensates for its deficiency in the CNS and preserves MNs [68,69]. However, because there is still no patient natural history available to validate the correlation between low SMN and the vulnerability of other organs beyond MNs, the systemic ASO delivery in a human trial is still under evaluation [65].

### 5.3. Gene Therapy for SMA: SMN1 Gene Replacement

Contrary to augmenting SMN production via targeting *SMN2*, another therapeutic approach aims to transfer the *SMN1* gene into the neural cells. Among various gene-delivery vectors, the self-complementary adeno-associated virus 9 (AAV9) was found to be the most promising because it is able to cross the BBB, and infected approximately 60% of MNs [70]. Besides, sustained human *SMN1* expression enhances rapid and efficient SMN production, which has been approved for in vivo study [71,72].

The first AAV9-*SMN1* gene therapy, Zolgensma (AVXS-101 or onasemnogene abeparvovec), was administered in 15 type 1 SMA patients, and showed a significant improvement in motor and survival in a phase 1/2a trial [39]. The follow-up study further verified a significant efficacy of early therapy [73]. The preliminary data from an ongoing open-label phase 3 trial of type 1 SMA infants continue to demonstrate promising results [74]. However, the FDA has recently placed a partial hold on intrathecal Zolgensma administration in older SMA patients (≥2 years and <5 years) based on safety concerns [75].

Because of different study designs, it is difficult to compare the efficacies between currently approved ASO-based and gene-based SMA therapies. However, the mean age of the patients was slightly lower in the Zolgensma trial than in the nusinersen trial (3.4 months vs. 5.4 months). There are several advantages of scAAV9-based gene therapy that make it potentially superior to ASO SMN-augmentation therapy [76]. First, scAAV9 gene therapy may require only a single intravenous infusion with a sustainable effect, whereas nusinersen probably requires lifelong repetitive intrathecal treatment. Second, given that SMN protein is ubiquitously expressed, systemic intravenous delivery of the AAV-vector gene has the advantage of increasing SMN expression in other organs in the body [6].

Nevertheless, there are still several concerns to be addressed, as independent studies with AAV9-*SMN* gene therapy administered through the intravenous route in large animals have shown suboptimal outcomes [77,78]. One of the notable concerns of AAV9-based gene therapy is the ubiquitous expression of SMN, leading to the nonspecific sequestration of essential RNAs and proteins through RNA–protein and protein–protein interactions, respectively. Furthermore, poor body-wide delivery of viral particles and likely immune response remains a concern for approaches based on gene replacement therapy [79].

### 5.4. Risdiplam

A small molecule agent, RG7800, was demonstrated to increase FL-*SMN2* transcripts in laboratory studies and human safety in a phase 1 trial of SMA patients [80]. However, dosing was suspended due to off-target retinal effects in long-term, nonclinical safety studies in monkeys [81]. Another modified RG7800-like compound, Risdiplam (RG7916), showed a comparative efficacy in in vitro and in vivo studies, as well as in human pharmacokinetic data [82]. The potential of distribution in both central and peripheral tissues makes Risdiplam a potent therapeutic agent for addressing SMA as a whole-body disease [83]. Both trials in type 1 SMA patients (FIREFISH) and in types 2 and 3 patients (SUNFISH) demonstrated not only a significant increment of SMN protein in the blood but also an improvement of motor function with event-free survival [84,85].

### 5.5. Branaplam

Branaplam (LMI070) can interact with U1 snRNP to facilitate exon 7 inclusion of SMN2 transcript, and thereby increases SMN protein levels and improves phenotypes [86]. An active phase ½ clinical trial of Branaplam is an open-label, first-in-human study with oral administration to evaluate the safety and efficacy in patients with type 1 SMA. The preliminary results showed significant improvement in the motor functions after 86 days of treatment. Five patients continued to improve after 127 days of treatment [87].

### 5.6. Celecoxib

Treatment with celecoxib, a cyclooxygenase 2 inhibitor, was shown to increase SMN in SMA cell and animal models [88]. Celecoxib has several advantages in treating SMA, including the ability to cross the BBB and favorable safety profiles in humans. A phase 2 trial in patients with SMA types 2 and 3 is actively recruiting patients [41]. It may be with the hope that celecoxib may serve as adjunctive therapy for SMA, particularly given the low safe doses required for SMN induction.

### 5.7. Quinazoline

Blocking of decapping scavenger enzyme (DcpS) has been shown to increase FL-*SMN2* transcript through upregulating *SMN2* promoter activity [89]. Quinazoline (Repligen or RG3039), a DcpS inhibitor, was demonstrated to increase SMN protein and survival in SMA mice [90]. However, a phase 1b trial showed that even though RG3039 successfully blocked DcpS in patients’ blood, the SMN protein level did not change significantly [91]. Therefore, the pharmaceutic company concluded that RG3039 would be ineffective in SMA patients, and the further trial was halted [92].

### 5.8. SMN Protein Stabilizers

Aminoglycoside antibiotics (from a class of FDA-approved drugs including tobramycin, geneticin, and amikacin) can mask premature stop codon mutations and promote read-through of exon 8, and thereby stabilize or increase the SMN level in patient fibroblasts [93,94]. However, they have only shown in vivo efficacy, and the toxicity has yet to be tested in animal models of SMA [46].

BBrm2 is a repurposed FDA-approved azithromycin acting on stop codon read-through, which was found to increase SMN in SMA patient cell lines and improve motor function and survival when intrathecally delivered in an SMA mouse model [46]. The pre-clinical trial was extended to another SMA mouse model, which also showed promising data [95].

Bortezomib is a ubiquitin proteasome inhibitor known to prevent SMN protein degradation. It has been shown to increase SMN protein in cultured cells and peripheral tissues of SMA model mice. Bortezomib-treated animals had improved motor function, which was associated with reduced spinal cord and muscle pathology and improved neuromuscular junction size, but no change in survival [96].

## 6. SMN-Independent Therapies for SMA

As the first SMN-dependent therapies are emerging into the clinical arena, other approaches beyond SMN augmentation are also under active investigation. However, SMN-dependent approaches pose a particular challenge for patients with chronic forms (types 3 and 4) of SMA, who are often diagnosed beyond the critical therapeutic window [97]. For the patients with the chronic form of SMA with a substantial loss of MNs, it is more crucial to target the SMN-independent pathways disrupted downstream of SMN. Furthermore, emerging evidence has substantiated that SMA is a systemic disorder that goes beyond motor neurons. Identifying non-SMN targets to develop combinatorial therapeutic approaches is tempting because a comprehensive whole-lifespan approach to SMA therapy is required that includes both SMN-dependent and SMN-independent strategies that treat the CNS and periphery together [6,98]. 

### 6.1. Neuroprotective Agents

Olesoxime (TRO19622) is a trophos cholesterol-oxime compound family of mitochondrial pore modulators with neuroprotective properties. Pre-clinical studies suggest that it improves the function and survival of neurons [99]. A phase 2 placebo-controlled trial in patients with types 2 and 3 SMA showed stabilized motor function at 24 months of treatment [100]. However, a subsequent follow-up study at 18 months did not demonstrate a significant clinical benefit (OLEOS, NCT02628743), and the pharmaceutical company announced that it was ending the development of olesoxime for SMA in June 2018 [6].

Other potentially neuroprotective agents, riluzole and gabapentin, have been investigated for their effects in treating SMA [101,102]. A phase 2/3 multicenter, randomized, double-blind study to assess the efficacy and safety of riluzole in young adults with types 2 and 3 SMA has been completed. Unfortunately, most of the results were not encouraging, or the studies were not adequate to show efficacy [1,41].

### 6.2. Myostatin Inhibitors

Recently, most of the SMN-independent therapies have focused on muscle, since muscle weakness is always prominent in SMA. Myostatin is a growth factor produced primarily in skeletal muscle cells to inhibit muscle growth. Theoretically, blocking the myostatin signaling pathway can induce increased muscle mass and consequently improve muscle strength and motor function [103]. Follistatin is an endogenous antagonist of myostatin, and over-expression of recombinant follistatin in SMA mouse muscle leads to increased skeletal muscle mass as well as survival [104]. On the other hand, inhibition of activin receptor type IIB (ActRIIB) ligands can promote muscle growth, which suggests a potential therapy for neuromuscular disorders, including SMA. The systemic delivery of AAV-mediated soluble inhibitor of ActRIIB showed improvements in both muscle mass and muscle function in the SMA mouse model [105]. BIIB 110 (ALG 801) is a recombinant inhibitor of ActRIIB, which is undergoing a phase 1a trial [106].

Another myostatin inhibitor, SRK-015 (Scholar Rock), is a human monoclonal antibody found to increase muscle mass in SMA mice [107]. A phase 2 trial in type 2 and type 3 SMA patients through monthly intravenous administration is underway, and the preliminary data demonstrated a robust and dose-dependent target engagement on myostatin precursor [108].

### 6.3. Skeletal Muscle Troponin Activator: Reldesemtiv

Reldesemtiv (CK-2127107) is a fast skeletal muscle troponin activator which has been shown to improve muscle function and physical performance in SMA [46]. Reldesemtiv was demonstrated to increase skeletal muscle force in response to nerve stimulation, associated with a calcium-sensitizing effect [109]. With promising results demonstrating prolonged stamina and a modest improvement in pulmonary function [110], a double-blind phase 2 trial is ongoing to examine the efficacy of oral administration twice a day in non-type 1 SMA patients [111].

### 6.4. Agents Targeting Neuromuscular Junction, Synapse, or Neurotransmitter

Pyridostigmine (Mestinon) is an anti-acetylcholinesterase drug approved for treating myasthenia gravis. Researchers believe that the medicine’s ability to activate and strengthen muscles might benefit SMA patients [112]. A placebo-controlled trial of pyridostigmine is ongoing to test the effects on muscle strength and fatigue in patients with types 2–4 SMA [113].

4-Aminopyridine (4-AP or Ampyra), a broad-spectrum inhibitor of potassium channels, is approved by the FDA for multiple sclerosis treatment. 4-AP was shown to improve the phenotypes of SMA in a *Drosophila* model, possibly through the pathway of motor circuits [114]. A phase 2/3 clinical trial assessing the efficacy in walking ability and endurance of type 3 adult SMA patients was completed in 2017, and the results are pending [115].

### 6.5. Stem Cell Therapy

The potential of cell therapy in SMA is related to the ability of stem cells to provide support to endogenous degenerating MNs [116]. Two currently available stem cell transplantation studies for SMA showed that primary neural stem cells injected into the spinal canal engrafted to the spinal cord, improved motor function, and extended survival [117,118]. However, these results have only reflected benefits likely with trophic support but without evidence of functional cell replacement. Accurate validation of therapeutic impact and a precise definition of the mechanism of action is still pending.

## 7. Combination Therapy for SMA

The concept that a combination of different therapeutic strategies could maximize the benefits for SMA treatment is intriguing. Although combined therapies with expensive drugs may at some point be prohibitive, limited data support the efficacy of such combinations on humans, and physicians and scientists are encouraged to explore all therapeutic possibilities [119]. Excitingly, a combined approach using SMN-dependent ASO-inducing *SMN2* exon inclusion and SMN-independent myostatin inhibition have shown a favorable result in an SMA animal model [120]. Combined treatment with Zolgensma and nusinersen has recently been investigated in a small group of patients, although the long-term benefit is still unclear [121]. Zolgensma and nusinersen have different mechanisms of action, so the drug-to-drug interaction is less likely. Nusinersen works by targeting an intron sequence to enhance exon 7 inclusion. However, a transferred gene of Zolgensma does not contain any introns, so its translation should not interfere with nusinersen [98]. Because thrombocytopenia has been reported as an adverse event in association with nusinersen, caution is required when Zolgensma treatment is considered. Longer-term follow-up data, especially in the treatment of pre-symptomatic patients, should be accumulated to assess the efficacy and risks of combination therapy.

## 8. Conclusions

Although two SMN-dependent therapies have entered the market, neither of them has been proved to provide a cure. Ongoing research is continuing to pursue more potential agents through the development of novel compounds. Each of the treatments mentioned above could be promising in treating SMA. However, we are facing a rapidly changing landscape in SMA because of the perspectives of novel therapeutics relying on a greater understanding of SMA pathomechanisms, as well more abundant knowledge of the natural history and the impact of wide-spread multidisciplinary care. With the increasing number of therapies for SMA, ethical and thoughtful consideration of all players (industry, the health-care system, patients, and caregivers) will soon be necessary. Support groups such as the SMA Foundation, CureSMA, and Fight SMA have played a vital role in research efforts, in addition to providing a community for families affected by SMA.

## Figures and Tables

**Figure 1 ijms-21-03297-f001:**
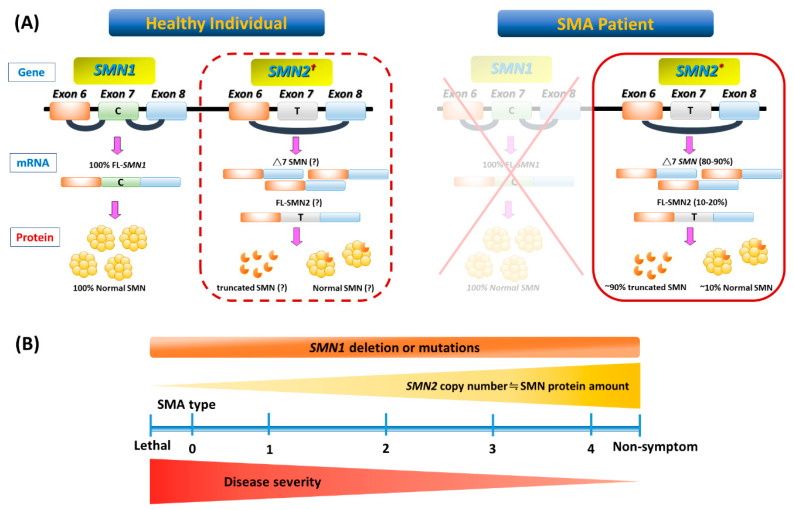
Genetic basis and phenotype-genotype correlation of spinal muscular atrophy (SMA). (**A**) In a healthy individual, full-length (FL) survival motor neuron (SMN) mRNA and protein arise from the *SMN1* gene. Patients with SMA have homozygous deletion or mutation of *SMN1* but retain at least one *SMN2* (indicated with an asterisk in the solid-border box on the right). However, *SMN2* can be dispensable in a healthy individual (indicated with an obelisk in the dotted-border box on the left). This single-nucleotide change in exon 7 (C-to-T) of *SMN2* causes alternative splicing during transcription, resulting in most *SMN2* mRNA lacking exon 7 (△7 SMN). About 90% of △7 SMN transcripts produce unstable truncated SMN protein, but a minority include exon 7 and code for FL, which maintains a degree of MN survival. (**B**) A continuous spectrum of phenotypes in SMA. Even with genetic confirmation of *SMN1* absence or mutations in all patients, SMA presentation ranges from very compromised neonates (type 0) to adults with minimal manifestations (type 4) depending on the *SMN2* numbers and FL SMN produced by each patient and modulated by potential disease modifiers that influence the final phenotype.

**Figure 2 ijms-21-03297-f002:**
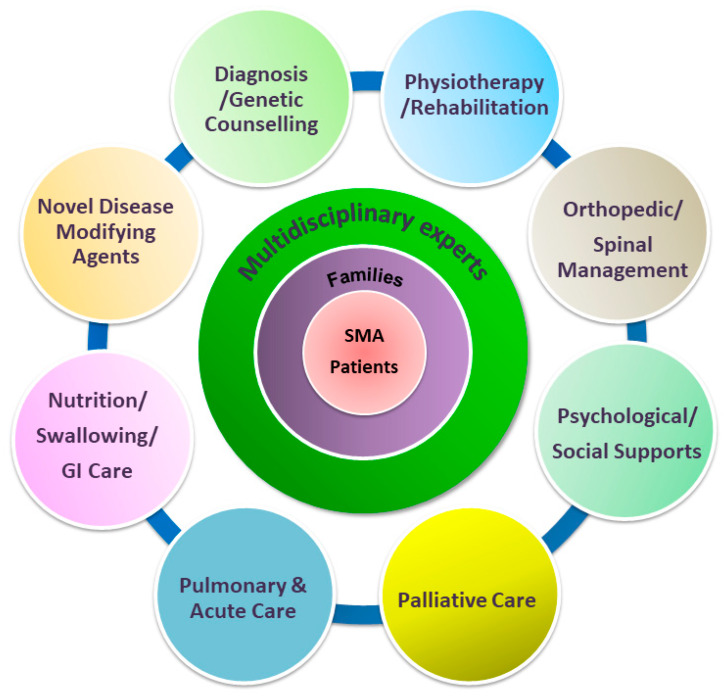
Paradigm of multidisciplinary care of SMA, incorporating disease-modifying therapies with supportive care. Novel disease-modifying medications and evolving multidisciplinary supportive management need to occur concomitantly to achieve the best possible outcome for SMA patients.

**Figure 3 ijms-21-03297-f003:**
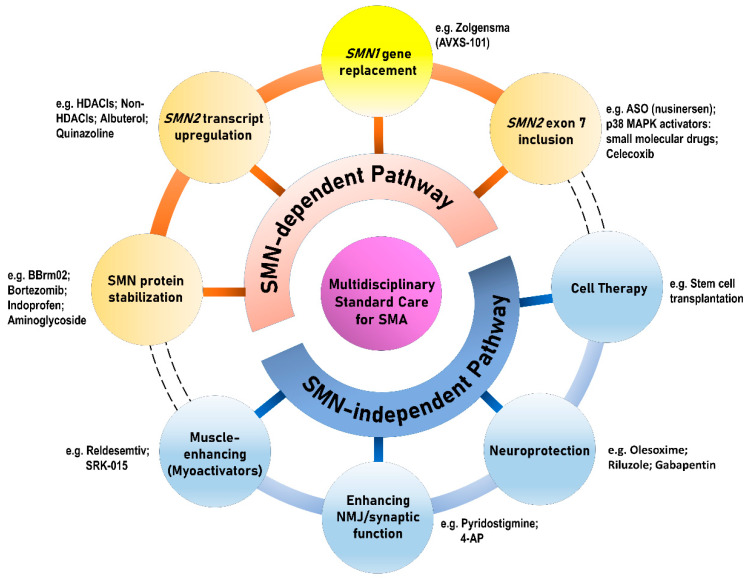
Therapeutic approaches for SMA. ASO: antisense oligonucleotide; HDACI: histone deacetylase inhibitor; NMJ: neuromuscular junction.

**Table 1 ijms-21-03297-t001:** Classification and subtypes of spinal muscular atrophy.

SMA Type (Historical Name)	OMIM	Onset Age	Motor Milestones Achieved	Subclassification	Natural History	Other Features	Estimated *SMN2* Copies	Estimated SMA Proportion
Type 0	-	Prenatal or at birth	Never sits, never head control	-	Death < 1 mo if untreated	Joint contractures, cardiac defect, facial diplegia, immediate respiratory failure after birth	1 *SMN2* copy in ~100% of patients	Unclear, Maybe < 1%
Type 1 (Werdnig-Hoffmann disease)	253300	0–6 mo	Never sits, some achieve head control	1A: Onset < 1 mo, usually by 2 wk; head control absent 1B: Onset 1–3 mo; poor or absent head control 1C: Onset 3–6 mo; head control achieved	1A: Death < 6 mo if untreated 1B and 1C: death < 2 yr if untreated	1A: Very similar to type 0 SMA 1B and 1C: Tongue fasciculation, swallowing difficulties, early respiratory failure	1 or 2 *SMN2* Copies in ~80% of patients	~60%
Type 2 (Dubowitz disease)	253550	7–18 mo	Sits but never stands	2A: Sits independently, may lose the ability to sit in later life 2B: Sits independently, maintains the ability to sit According to functional level, decimal classification ranging from 2.1 to 2.9	Usually survive >2 yr; ~70% alive at 25 yr	Proximal weakness, postural hand tremor, normal intellectual ability, kyphoscoliosis	3 *SMN2* copies in >70% patients	~27%
Type 3 (Kugelberg–Welander disease)	253400	>18 mo	Stands and walks	3A: Onset between 18 and 36 mo 3B: Onset >3 yr	Survival into adulthood	May have hand tremor, resembles muscular dystrophy 3A: Scoliosis, usually early loss of ambulation	3 or 4 *SMN2* copies in ~95% of patients	~12%
Type 4	271150	10–30 yr, usually >21 yr	Stands and walks	-	Survival into adulthood	Usually preserved walking ability	4 or more *SMN2* copies in >90%	~1%

SMA: spinal muscular atrophy; mo: months; yr: years.

**Table 2 ijms-21-03297-t002:** Novel therapeutic approaches in spinal muscular atrophy: current clinical and preclinical trials.

Therapeutic Pathways	Pathologic Points	Therapeutic Targets	Therapeutic Agents	Trial Status (Completed or Ongoing)/Results
SMN-dependent	*SMN1* mutation	*SMN1* replacement	Zolgensma (AVXS-101)	FDA-Approved
	Alternative splicing of *SMN2* mRNA	Promote exon 7 inclusion	Nusinersen (Spinraza) Risdiplam (RG7916) Branaplam (LMI070)	Nusinersen: FDA-approved Risdiplam: ongoing phase 2/3 placebo-controlled; approaching FDA-approved Branaplam: ongoing phase 1/2 open-label
	Decreased full length SMN mRNA	Upregulation of *SMN2* transcript	HDACIs, e.g., PBA, VPA, Non-HDACIs: Hydroxyurea Celecoxib Quinazoline (RG3039) Albuterol Prolactin	PBA: completed placebo-controlled; negative VPA: completed placebo-controlled; negative Hydroxyurea: completed placebo-controlled; negative Celecoxib: ongoing phase 2 open-label Quinazoline: suspended Albuterol: completed open-label; positive but lacking large controlled trials data Prolactin: preclinical
	SMN protein degradation	Stabilizing SMN protein	Aminoglycoside Bortezomib BBrm02 Indoprofen polyphenols	All are preclinical
SMN-independent	Anabolic abnormalities	Muscle-enhancing agent (Myoactivators)	SRK-015 Reldesemtiv (CK-2127107) BIIB110 (ALG 801) Follistatin	SRK-015: ongoing phase 2 open-label Reldesemtiv: completed phase 2 placebo-controlled; pending BIIB110: ongoing phase 1a Follistatin: preclinical
	Neuromuscular junction defect	Enhancing neurotransmitters	Pyridostigmine (Mestinon) 4-aminopyridine (4-AP)	Pyridostigmine: completed placebo-controlled trial; pending 4-aminopyridine: completed placebo-controlled trial; pending
	Motor neuron loss	Neuroprotection	Riluzole Gabapentin Olesoxime (TRO19622)	Riluzole: completed open-label; negative Gabapentin: placebo-controlled trial; negative Olesoxime: suspended
		Cell therapy for neurotrophic support	Stem cells	Preclinical

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
