# Peer review of "New and Developing Therapies in Spinal Muscular Atrophy: From Genotype to Phenotype to Treatment and Where Do We Stand?"

_ijms, 2020, doi:10.3390/ijms21093297_

Round 1

Reviewer 1 Report

This review covers the pathomechanism and genotype phenotype of SMA to the current status of therapeutic development. It is considered to be effective in understanding the current and future of SMA medication. I have no specific objections to the content of the text. However, some modification to the references are required.

  1. There are some errors in the description of references (page).Ref 11, 61
  2. It seems better to list references for cardiac defects
  3. I have questions about some references

Ref: 17 is not the main article on respiratory care and nutritional management for advanced cases

Ref 26 is not a paper of standard-of-care

Ref 50 is a commentary to the paper written by Kaiya S, et al (J Clin Invest 2014; 124: 785-800).

Author Response

There are some errors in the description of references (page). Ref 11, 61

Reply: We appreciate Reviewer 1’s valuable comments. We have relocated the Ref 11 for more appropriate citation and deleted Ref 61.

It seems better to list references for cardiac defects

Reply: We have added the description and reference regarding congenital cardiac defects in severe SMA types 0 and 1 in the revised manuscript (Page 9, Line 10-12).

I have questions about some references

Ref: 17 is not the main article on respiratory care and nutritional management for advanced cases

Reply: We have deleted the previous Ref 17 and add relevant data of references regarding respiratory care and nutritional management for advanced cases with type 2 SMA (Page 10, Line 4, Ref 21&22).

Ref 26 is not a paper of standard-of-care

Reply: We have replaced with an update reference regarding standard-of-care (Page 12, Line 5, Ref 21&22)

Ref 50 is a commentary to the paper written by Kariya S, et al (J Clin Invest 2014; 124: 785-800).

Reply: We have replaced with the original paper by Kariya S, et al. (Page 20, Line 3, Ref 59).

Reviewer 2 Report

The authors describe new treatments for SMA, review the current therapies and try to make a correlation between genotype and phenotype.

Although the paper tries to  add new information to the field, there are a major concerns:

1 they have missed other players in SMA,as glial cells and they have not mention neuroinflammation  ( Abati et al, Neurobiology of disease,2020);

2 they should mention for completeness  the SMARD 1 ( spinal muscular dystrophy type 1 with respiratory distress ( Perego et al , Cell mol life Sci,2020);

3They should mention longitudinal studies Mercuri et al Orphanet Rare Dis,2020;

#4 the references are not updated;

Author Response

Although the paper tries to add new information to the field, there are a major concerns:

  1. They have missed other players in SMA, as glial cells and they have not mention neuroinflammation ( Abati et al, Neurobiology of disease,2020)

Reply: We appreciate Reviewer 2’s valuable comments. We also think neuroinflammation might also play an essential role in SMA pathomechanism. We have added the description and references regarding the role of glial cells and possible neuroinflammatory mechanisms in SMA pathogenesis (Page 7, Line 2-6, & Ref 14, 15).

  1. They should mention for completeness the SMARD 1 ( spinal muscular dystrophy type 1 with respiratory distress ( Perego et al , Cell mol life Sci,2020)

Reply: Thanks for Review 2’s recommendation. We have added the insights regarding therapeutic approaches in SMARD1 in the revised manuscript (Page 15, Line 13-18 & Ref 42).

  1. They should mention longitudinal studies Mercuri et al Orphanet Rare Dis,2020.

Reply: Thanks for Reviewer 2 suggesting an update reference for SMA natural history. We had added this reference in the revised manuscript (Page 10, Line 18 & Ref 26).

  1. The references are not updated.

Reply: We have updated references to become up-to-date, and the majority of references to the manuscripts are published within five years.

Reviewer 3 Report

The overall structure of the review is clear and sound. The author reviewed most of the current therapies (clinical trials) to treat SMA, with discussions of the pro and cons of some of the strategy. This will give readers a clear overview of the current status of the SMA treatment. Please see below some comments:  

  1. In page 4, line 65, the author claims that:

“However, such a phenotype-genotype correlation is not so absolute, as recent studies have indicated that additional cellular mechanisms, like positive or negative disease modifiers, might also involve the modulation of SMA clinical severity [4,5].”

It would be clearer and helpful to explain the work done in ref 4 and 5 to give clues of what disease modifiers have been reported to affect the SMA clinical severity.

  1. In page 5, line 77, it is written: “However, the multifaceted roles of SMN protein are still under investigation, and it is unclear how a deficiency in ubiquitously expressed SMN can selectively cause the dramatic MN degeneration. The cell-autonomous effects related to deficient SMN are responsible for the MNs degeneration; however, it does not account for the full SMA phenotype, implicating not only dysfunction of neural networks but other cell types involved in the disease process [11,12].”

And this opinion is emphasised again in page 14, line 196: “However, increasing evidence extend the pathogenic effect of SMN deficiency beyond MNs to include additional cells both within and outside the CNS, whereby numerous peripheral organs and tissues demonstrate pathological changes both in pre-clinical models and patients [11,24].”

Indeed more and more studies have suggested that SMA is not a simple disease affect only MN, but more of a complicated disease that multi-organs are affected.  Could author include works done about the involvement of other organs such as vascular system or intestine system to make it clear?

  1. Page 26, line 431: Does  Activin Receptor Type IIB Inhibitor also work along myostatin pathway? If so, this section should be included in the myostatin inhibitor section together with Follistatin.

  1. Page 29, line 453: “primary MN stem cells” should be “primary neural stem cells”.

  1. Page 29, line 464: “Excitingly, combined SMN-dependent and SMN-independent therapeutic agents on a SMA animal model have shown a favorable result [111].” This work was performed using the combination of AON to induce the SMN2 exon inclusion and the myostatin inhibition strategy; it would be nice to explain here.

Author Response

In page 4, line 65, the author claims that:

“However, such a phenotype-genotype correlation is not so absolute, as recent studies have indicated that additional cellular mechanisms, like positive or negative disease modifiers, might also involve the modulation of SMA clinical severity [4,5].”

It would be clearer and helpful to explain the work done in ref 4 and 5 to give clues of what disease modifiers have been reported to affect the SMA clinical severity.

Reply: We appreciate Reviewer 3’s valuable comments. We have added the description of the update work done by Ref 4 and 5 and listed the reported SMA modifiers to make the readers more easily to realize (Page 6, Line 4-6 & added Ref 4).

In page 5, line 77, it is written: “However, the multifaceted roles of SMN protein are still under investigation, and it is unclear how a deficiency in ubiquitously expressed SMN can selectively cause the dramatic MN degeneration. The cell-autonomous effects related to deficient SMN are responsible for the MNs degeneration; however, it does not account for the full SMA phenotype, implicating not only dysfunction of neural networks but other cell types involved in the disease process [11,12].”

And this opinion is emphasised again in page 14, line 196: “However, increasing evidence extend the pathogenic effect of SMN deficiency beyond MNs to include additional cells both within and outside the CNS, whereby numerous peripheral organs and tissues demonstrate pathological changes both in pre-clinical models and patients [11,24].”

Indeed more and more studies have suggested that SMA is not a simple disease affect only MN, but more of a complicated disease that multi-organs are affected. Could author include works done about the involvement of other organs such as vascular system or intestine system to make it clear?

Reply: As Reviewer 3’s suggestion, we have included works and more references regarding the involvement of other organs beyond motor neurons in SMA phenotypes, making it more clear to elucidate multi-organs involvement in SMA pathogenesis (Pages 14 ,Line 16-18; Page 15, Line 1-2, Ref 12,30,35-38)

Page 26, line 431: Does Activin Receptor Type IIB Inhibitor also work along the myostatin pathway? If so, this section should be included in the myostatin inhibitor section together with Follistatin.

Reply: As Activin Receptor Type IIB Inhibitor also works along the myostatin pathway, we have moved the description of BIIB 110 (ALG 801) to the myostatin inhibitor section (Page 28, Line 1-7).

Page 29, line 453: “primary MN stem cells” should be “primary neural stem cells”.

Reply: Thanks for Reviewer 3’s comment, and we have corrected the term (Page 29, Line 18).

Page 29, line 464: “Excitingly, combined SMN-dependent and SMN-independent therapeutic agents on a SMA animal model have shown a favorable result [111].” This work was performed using the combination of AON to induce the SMN2 exon inclusion and the myostatin inhibition strategy; it would be nice to explain here.

Reply: Thanks for Reviewer 3’s suggestion. We have added this description in the sentence regarding the combinatory therapy for SMA (Page 30, Line 12-14).

Round 2

Reviewer 2 Report

The paper is now very much improved and the authors have addressed all points raised by Referees.

Author Response

Dear Reviewer 2,

Thanks for your valuable comments on our manuscript.

Best regards,

Tai-Heng Chen